# Neural Conservation Laws:
# A Divergence-Free Perspective

**Jack Richter-Powell**
Vector Institute
jack.richter-powell@mcgill.ca

**Yaron Lipman**
Meta AI
ylipman@meta.com

**Ricky T. Q. Chen**
Meta AI
rtqichen@meta.com

## Abstract

We investigate the parameterization of deep neural networks that by design satisfy the continuity equation, a fundamental conservation law. This is enabled by the observation that any solution of the continuity equation can be represented as a divergence-free vector field. We hence propose building divergence-free neural networks through the concept of differential forms, and with the aid of automatic differentiation, realize two practical constructions. As a result, we can parameterize pairs of densities and vector fields that always exactly satisfy the continuity equation, foregoing the need for extra penalty methods or expensive numerical simulation. Furthermore, we prove these models are universal and so can be used to represent any divergence-free vector field. Finally, we experimentally validate our approaches by computing neural network-based solutions to fluid equations, solving for the Hodge decomposition, and learning dynamical optimal transport maps.

## 1 Introduction

Modern successes in deep learning are often attributed to the expressiveness of black-box neural networks. These models are known to be universal function approximators [Hornik et al., 1989]—but this flexibility comes at a cost. In contrast to other parametric function approximators such as Finite Elements [Schroeder and Lube, 2017], it is hard to *bake* exact constraints into neural networks. Existing approaches often resort to penalty methods to approximately satisfy constraints—but these increase the cost of training and can produce inaccuracies in downstream applications when the constraints are not exactly satisfied. For the same reason, theoretical analysis of soft-constrained models also becomes more difficult. On the other hand, enforcing hard constraints on the architecture can be challenging, and even once enforced, it is often unclear whether the model remains sufficiently expressive within the constrained function class.

In this work, we discuss an approach to directly bake in two constraints into deep neural networks: (i) **having a divergence of zero**, and (ii) **exactly satisfying the continuity equation**. One of our key insights is that *the former directly leads into the latter*, so the first portion of the paper focuses on divergence-free vector fields. These represent a special class of vector fields which have widespread use in the physical sciences. In computational fluid dynamics, divergence-free vector fields are used to model incompressible fluid interactions formalized by the Euler or Navier-Stokes equations. In $\mathbb{R}^3$ we know that the curl of a vector field has a divergence of zero, which has seen many uses in graphics simulations (*e.g.*, Eisenberger et al. [2018]). Perhaps less well-known is that lurking behind this fact is the generalization of divergence and curl through differential forms [Cartan, 1899], and the powerful identity $d^2 = 0$. We first explore this generalization, then discuss two constructions derived from it for parmaterizing divergence-free vector fields. While this approach has been partially discussed previously [Barbarosie, 2011, Kelliher, 2021], it is not extensively known and to the best of our knowledge has not been explored by the machine learning community.

36th Conference on Neural Information Processing Systems (NeurIPS 2022).

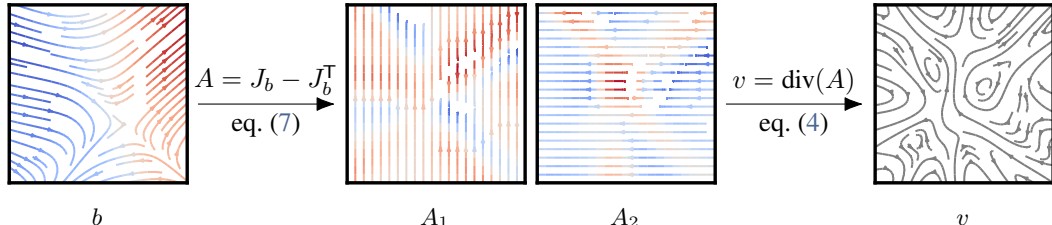

Figure 1: Divergence-free vector fields $v : \mathbb{R}^d \to \mathbb{R}^d$ can be constructed from an antisymmetric matrix field $A : \mathbb{R}^d \to \mathbb{R}^{d \times d}$ or an arbitrary vector field $b : \mathbb{R}^d \to \mathbb{R}^d$. $J_b$ represents the Jacobian matrix of $b$, and $A_1$ and $A_2$ are the first and second rows of $A$. Color denotes divergence.

Concretely, we derive two approaches—visualized in Figure 1—to transform sufficiently smooth neural networks into divergence-free neural networks, made efficient by automatic differentiation and recent advances in vectorized and composable transformations [Bradbury et al., 2018, Horace He, 2021]. Furthermore, both approaches can theoretically represent any divergence-free vector field.

We combine these new modeling tools with the observation that solutions of the continuity equation—a partial differential equation describing the evolution of a *density* under a *flow* —can be characterized jointly as a divergence-free vector field. As a result, we can parameterize neural networks that, by design, always satisfy the continuity equation, which we coin Neural Conservation Laws (NCL). While prior works either resorted to penalizing errors [Raissi et al., 2019] or numerically simulating the density given a flow [Chen et al., 2018], baking this constraint directly into the model allows us to forego extra penalty terms and expensive numerical simulations.

## 2   Constructing divergence-free vector fields

We will use the notation of *differential forms* in $\mathbb{R}^n$ for deriving the divergence-free and universality properties of our vector field constructions. We provide a concise overview of differential forms in Appendix A. For a more extensive introduction see *e.g.*, Do Carmo [1998], Morita [2001]. Without this formalism, it is difficult to show either of these two properties; however, readers who wish to skip the derivations can jump to the matrix formulations in equation 4 and equation 7.

Let us denote $\mathcal{A}^k(\mathbb{R}^n)$ as the space of differential $k$-forms, $d : \mathcal{A}^k(\mathbb{R}^n) \to \mathcal{A}^{k+1}(\mathbb{R}^n)$ as the exterior derivative, and $\star : \mathcal{A}^k(\mathbb{R}^n) \to \mathcal{A}^{n-k}(\mathbb{R}^n)$ as the Hodge star operator that maps each $k$-form to an $(n\text{-}k)$-form. Identifying a vector field $v : \mathbb{R}^d \to \mathbb{R}^d$ as a 1-form, $v = \sum_{i=1}^n v_i dx_i$, we can express the divergence $\text{div}(v)$ as the composition $d \star v$.

To parameterize a divergence-free vector field $v : \mathbb{R}^n \to \mathbb{R}^n$, we note that by the fundamental property of the exterior derivative, taking an arbitrary $(n\text{-}2)$-form $\mu \in \mathcal{A}^{n-2}(\mathbb{R}^n)$ we have that

$$0 = d^2\mu = d(d\mu) \tag{1}$$

and since $\star$ is its own inverse up to a sign, it follows that

$$v = \star d\mu \tag{2}$$

is divergence free. We write the parameterization $v = \star d\mu$ explicitly in coordinates. Since a basis for $\mathcal{A}^{n-2}(\mathbb{R}^n)$ can be chosen to be $\star(dx_i \wedge dx_j)$, we can write $\mu = \frac{1}{2}\sum_{i,j=1}^n \mu_{ij} \star (dx_i \wedge dx_j)$, where $\mu_{ji} = -\mu_{ij}$ Now a direct calculation shows that up to a constant sign (see Appendix-A.1)

$$\star d\mu = \sum_{i=1}^n \left[ \sum_{j=1}^n \frac{\partial \mu_{ij}}{\partial x_j} \right] dx_i. \tag{3}$$

This formula is suggestive of a simple matrix formulation: If we let $A : \mathbb{R}^n \to \mathbb{R}^{n \times n}$ be the anti-symmetric matrix-valued function where $A_{ij} = \mu_{ij}$ then the divergence-free vector field $v = \star d\mu$ can be written as taking row-wise divergence of $A$, *i.e.*,

$$v = \begin{pmatrix} \text{div}(A_1) \\ \vdots \\ \text{div}(A_n) \end{pmatrix}. \tag{4}$$

However, this requires parameterizing $O(n^2)$ functions. A more compact representation, which starts from a vector field, can also be derived. The idea behind this second construction is to model $\mu$ instead as $\mu = \delta\nu$, where $\nu \in \mathcal{A}^{n-1}(\mathbb{R}^n)$. Putting this together with equation 2 we get that

$$v = \star d\delta\nu \tag{5}$$

is a divergence-free vector field. To provide equation 5 in matrix formulation we first write $\nu \in \mathcal{A}^{n-1}(\mathbb{R}^n)$ in the (n-1)-form basis, *i.e.*, $\nu = \sum_{i=1}^n \nu_i \star dx_i$. Then a direct calculation provides up to a constant sign

$$\delta\nu = \frac{1}{2} \sum_{i,j=1}^n \left[ \frac{\partial \nu_i}{\partial x_j} - \frac{\partial \nu_j}{\partial x_i} \right] \star (dx_i \wedge dx_j) \tag{6}$$

So, given an arbitrary vector field $b : \mathbb{R}^n \to \mathbb{R}^n$, and denoting $J_b$ as the Jacobian of $b$, we can construct $A$ as

$$A = J_b - J_b^\top. \tag{7}$$

where $J_b$ denotes the Jacobian of $b$.

To summarize, we have two constructions for divergence-free vector fields $v$:

**Matrix-field**: (equations 2 and 4) $v$ is represented using an anti-symmetric matrix field $A$.

**Vector-field**: (equations 5 and 4+7) $v$ is represented using a vector field $b$.

As we show in the next section, these two approaches are maximally expressive (*i.e.*, universal), meaning they can approximate arbitrary smooth divergence-free vector fields. However, empirically these two constructions can exhibit different practical trade-offs. The matrix-field construction has a computational advantage as it requires one less Jacobian computation, though it requires more components— $O(n^2)$ vs. $O(n)$—to represent than the vector-field construction. This generally isn't a concern as all components can be computed in parallel. However, the vector-field construction can make it easy to

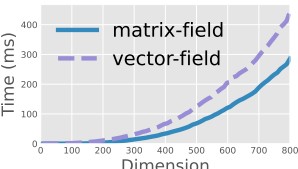

Figure 2: Compute times.

bake in additional constraints; an example of this is discussed in Section 7.1, where a non-negativity constraint is imposed for modeling continuous-time probability density functions. In Figure 2 we show wallclock times for evaluating the divergence-free vector field based on our two constructions. Both exhibit quadratic scaling (in function evaluations) with the number of dimensions due to the row-wise divergence in equation 4, while the vector-field construction has an extra Jacobian computation.

## 2.1 Universality

Both the matrix-field and vector-field representations are universal, *i.e.* they can model arbitrary divergence-free vector fields. The main tool in the proof is the Hodge decomposition theorem [Morita, 2001, Berger, 2003]. For simplicity we will be working on a periodic piece of $\mathbb{R}^n$, namely the torus $\mathbb{T}^n = [-M, M]^n / \sim$ where $\sim$ means identifying opposite edges of the $n$-dimensional cube, and $M > 0$ is arbitrary. Vector fields with compact support can always be encapsulated in $\mathbb{T}^n$ with $M > 0$ sufficiently large. As $\mathbb{T}^n$ is locally identical to $\mathbb{R}^n$, all the previous definitions and constructions hold.

**Theorem 2.1.** *The matrix and vector-field representations are universal in $\mathbb{T}$, possibly only missing a constant vector field.*

A formal proof of this result is in Appendix B.1.

## 3 Neural Conservation Laws

We now discuss a key aspect of our work, which is parameterizing solutions of scalar conservation laws. Conservation laws are a focal point of mathematical physics and have seen applications in machine learning, with the most well known examples being conserved scalar quantities often referred to as density, energy, or mass. Formally, a conservation law can be expressed as a first-order PDE

written in divergence form as $\frac{\partial \rho}{\partial t} + \text{div}(j) = 0$, where $j$ is known as the flux, and the divergence is taken over spatial variables. In the case of the continuity equation, there is a velocity field $u$ which describes the flow and the flux is equal to the density times the velocity field:

$$\frac{\partial \rho}{\partial t} + \text{div}(\rho u) = 0 \tag{8}$$

where $\rho : \mathbb{R}^n \to \mathbb{R}^+$ and $u : \mathbb{R}^n \to \mathbb{R}^n$. One can then interpret the equation to mean that $u$ transports $\rho(0, \cdot)$ to $\rho(t, \cdot)$ continuously – without teleporting, creating or destroying mass. Such an interpretation plays a key role in physics simulations as well as the dynamic formulation of optimal transport [Benamou and Brenier, 2000]. In machine learning, the continuity equation appears in continuous normalizing flows [Chen et al., 2018]—which also have been used to approximate solutions to dynamical optimal transport. [Finlay et al., 2020, Tong et al., 2020, Onken et al., 2021] These, however, only model the velocity $u$ and rely on numerical simulation to solve for the density $\rho$, which can be costly and time-consuming.

Instead, we observe that equation 8 can be expressed as a divergence-free vector field by augmenting the spatial dimensions with the time variable, resulting in a vector field $v$ that takes as input $(t, x)$ and outputs $(\rho, \rho u)$. Then equation 8 is equivalent to

$$\text{div}(v) = \text{div}\begin{pmatrix} \rho \\ \rho u \end{pmatrix} = \frac{\partial \rho}{\partial t} + \text{div}(\rho u) = 0 \tag{9}$$

where the divergence operator is now taken with respect to the joint system $(t, x)$, *i.e.* $\frac{\partial}{\partial t} + \sum_{i=1}^{n} \frac{\partial}{\partial x_i}$. We thus propose modeling solutions of conservation laws by parameterizing the divergence-free vector field $v$. Specifically, we parameterize a divergence-free vector field $v$ and set $v_1 = \rho$ and $v_{2:n+1} = \rho u$, allowing us to recover the vector field as $u = \frac{v_{2:n+1}}{\rho}$, assuming $\rho \neq 0$. This allows us to enforce the continuity equation at an architecture level. Compared to simulation-based modeling approaches, we completely forego such computationally expensive simulation procedures. Code for our experiments are available at `https://github.com/facebookresearch/neural-conservation-law`.

## 4 Related Works

**Baking in constraints in deep learning** Existing approaches to enforcing constraints in deep neural networks can induce constraints on the derivatives, such as convexity [Amos et al., 2017] or Lipschitz continuity [Miyato et al., 2018]. More complicated formulations involve solving numerical problems such as using solutions of convex optimization problems [Amos and Kolter, 2017], solutions of fixed-points iterations [Bai et al., 2019], and solutions of ordinary differential equations [Chen et al., 2018]. These models can help provide more efficient approaches or alternative algorithms for handling downstream applications such as constructing flexible density models [Chen et al., 2019, Lu et al., 2021] or approximating optimal transport paths [Tong et al., 2020, Makkuva et al., 2020, Onken et al., 2021]. However, in many cases, there is a need to solve a numerical problem in which the solution may only be approximated up to some numerical accuracy; for instance, the need to compute the density flowing through a vector field under the continuity equation [Chen et al., 2018].

**Applications of differential forms** Differential forms and more generally, differential geometry, have been previously applied in manifold learning—see *e.g.* Arvanitidis et al. [2021] and Bronstein et al. [2017] for an in-depth overview. Most of the applications thus far have been restricted to 2 or 3 dimensions—either using identities like div $\circ$ curl $= 0$ in 3D for fluid simulations [Rao et al., 2020], or for learning geometric invariances in 2D images or 3D space [Gerken et al., 2021, Li et al., 2021].

**Conservation Laws in Machine Learning** [Sturm and Wexler, 2022] previously explored discrete analogs of conservation laws by conserving mass via a balancing operation in the last layer of a neural network. [Müller, 2022] utilizes a wonderful application of Noether's theorem to model conservation laws by enforcing symmetries in a Lagrangian represented by a neural network.

## 5 Neural approximations to PDE solutions

As a demonstration of our method, we apply it to neural-based PDE simulations of fluid dynamics. First, we apply it to modelling inviscid fluid flow in the open ball $\mathbb{B} \subseteq \mathbb{R}^3$ with free slip boundary

conditions, then to a 2d example on the flat Torus $\mathbb{T}^2$, but with more complex initial conditions. While these are toy examples, they demonstrate the value of our method in comparison to existing approaches—namely that we can exactly satisfy the continuity equation and preserve exact mass.

**The Euler equations of incompressible flow**   The incompressible Euler equations [Feynman et al., 1989] form an idealized model of inviscid fluid flow, governed by the system of partial differential equations[1]

$$\frac{\partial \rho}{\partial t} + \text{div}(\rho u) = 0, \qquad \frac{\partial u}{\partial t} + \nabla_u u = \frac{\nabla p}{\rho}, \qquad \text{div}(u) = 0 \tag{10}$$

in three unknowns: the fluid velocity $u(t, x) \in \mathbb{R}^3$, the pressure $p(t, x)$, and the fluid density $\rho(t, x)$. While the fluid velocity and density are usually given at $t = 0$, the initial pressure is not required. Typically, on a bounded domain $\Omega \subseteq \mathbb{R}^n$, these are supplemented by the *free-slip* boundary condition and initial conditions

$$u \cdot n = 0 \text{ on } \partial\Omega \qquad u(0, x) = u_0 \text{ and } \rho(0, x) = \rho_0 \qquad \text{on } \Omega \tag{11}$$

The density $\rho$ plays a critical role since in addition to being a conserved quantity, it influences the dynamics of the fluid evolution over time. In numerical simulations, satisfying the continuity equation as closely as possible is desirable since the equations in (10) are coupled. Error in the density feeds into error in the velocity and then back into the density over time. In the finite element literature, a great deal of effort has been put towards developing conservative schemes that preserve mass (or energy in the more general compressible case)—see Guermond and Quartapelle [2000] and the introduction of Almgren et al. [1998] for an overview. But since the application of physics informed neural networks (PINNs) to fluid problems is much newer, conservative constraints have only been incorporated as penalty terms into the loss [Mao et al., 2020, Jin et al., 2021].

## 5.1   Physics informed neural networks

Physics Informed Neural Networks (PINNs; Raissi et al. [2019, 2017]) have recently received renewed attention as an application of deep neural networks. While using neural networks as approximate solutions to PDE had been previously explored (e.g in Lagaris et al. [1998]), modern advances in automatic differentiation algorithms have made the application to much more complex problems feasible [Raissi et al., 2019]. The "physics" in the name is derived from the incorporation of *physical* terms into the loss function, which consist of adding the squared residual norm of a PDE. For example, to train a neural network $\phi = [\rho, p, u]$ to satisfy the Euler equations, the standard choice of loss to fit to is

$$L_F = \left\| u_t + \nabla_u u + \frac{\nabla p}{\rho} \right\|_\Omega^2 \quad L_{\text{div}} = \|\text{div}(u)\|_\Omega \quad L_I = \|u(0, \cdot) - u_0(\cdot)\|_\Omega^2 + \|\rho(0, \cdot) - \rho_0(\cdot)\|_\Omega$$

$$L_{\text{Cont}} = \left\| \frac{\partial \rho}{\partial t} + \text{div}(\rho u) \right\|_\Omega^2 \qquad L_G = \|u \cdot n\|_{\partial\Omega}^2 \qquad L_{\text{total}} = \gamma \cdot [L_F, L_I, L_{\text{div}}, L_{\text{Cont}}, L_G]$$

where $\gamma = (\gamma_F, \gamma_I, \gamma_{\text{div}} \gamma_{Cont}, \gamma_G)$ denotes suitable coefficients (hyperparameters). The loss term $L_G$ ensures fluid does not pass through boundaries, when they are present. Similar approaches were taken in [Mao et al., 2020] and [Jagtap et al., 2020] for related equations. While schemes of this nature are very easy to implement, they have the drawback that since PDE terms are only penalized and not strictly enforced, one cannot make guarantees as to the properties of the solution.

To showcase the ability of our method to model conservation laws, we will parameterize the density and vector field as $v = [\rho, \rho u]$, as detailed in Section 3. This means we can omit the term $L_{\text{Cont}}$ as described in Section 5.1 from the training loss. The divergence penalty, $L_{\text{div}}$ remains when modeling incompressible fluids, since $u$ is not necessarily itself divergence-free – it is $v = [\rho, \rho u]$ which is divergence free. In order to stablize training, we can modify the loss terms $L_F, L_G, L_I$ to avoid division by $\rho$. This is detailed in Appendix B.2.

## 5.2   Incompressible variable density inside the 3D unit ball

We first construct a simple example within a bounded domain, specifically, we will consider the Euler equations inside $B(0, 1) \subseteq \mathbb{R}^3$, with the initial conditions

$$\rho(0, x) = 3/2 - \|x\|^2 \qquad v(0, x) = (-2, x_0 - 1, 1/2) \tag{12}$$

---

[1]The convective derivative appearing in equation 10, $\nabla_u u(x) = \lim_{h \to 0} \frac{u(x + hu(x)) - u(x)}{h} = [Du](u)$ is also often written as $(\nabla \cdot u)u$.

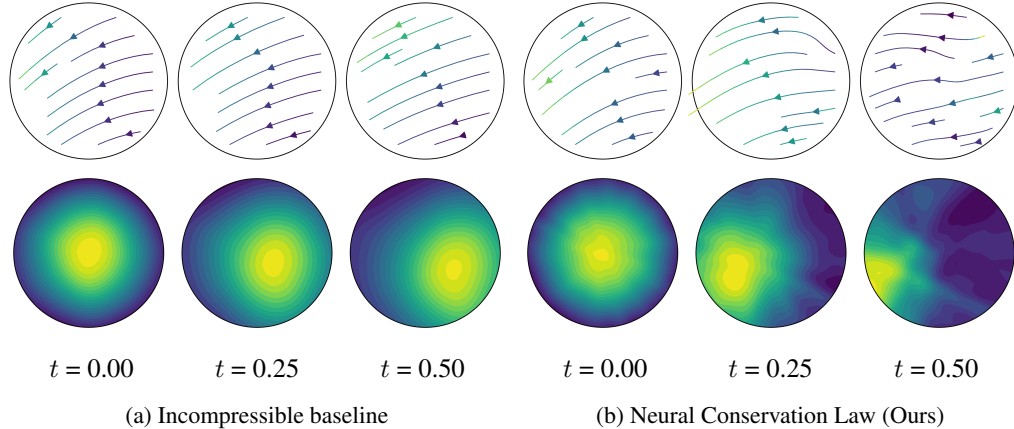



$t = 0.00$     $t = 0.25$     $t = 0.50$       $t = 0.00$     $t = 0.25$     $t = 0.50$

(a) Incompressible baseline       (b) Neural Conservation Law (Ours)



Figure 3: Exactly satisfying the continuity equation yields much better performance than only satisfying the incompressible (divergence-free) condition of the vector field, even though both approaches obtain very small loss values. 2D slices of the learned solution of $u$ and $\rho$ are shown.

As a baseline, we compare against a parameterization that uses the divergence-free vector field only for the incompressible constraint. This is equivalent to the curl approach described in Rao et al. [2020]; another neural network is used to parameterize $\rho$. In contrast, our generalized formulation of divergence-free fields allow us to model the continuity equation exactly, as in this case, $[\rho, \rho u]$ is 4 dimensional. We parameterize our method using $v = [\rho, \rho u]$, as detailed in Section 3, with the vector-field approach from Section 2.

The results for training both networks are shown in Figure 3. Surprisingly, even though both architectures signficantly improve the training loss after a comparable number of iterations (see Appendix C), the curl approach fails to advect the density accurately, whereas our NCL approach evolves the density as it should given the initial conditions.

## 5.3 Incompressible variable density flow on $\mathbb{T}^2$

As a more involved experiment, we model the Euler equations on the flat 2-Torus $\mathbb{T}^2$. This is parameterized by a multilayer perceptron pre-composed with the periodic embedding $\iota : [0, 1]^2 \to \mathbb{T}^2$

$$\iota(x, y) = [\cos(2\pi x), \sin(2\pi x), \cos(2\pi y), \sin(2\pi y)]$$

The continuity equation is enforced exactly, using the method detailed in Section 3, so the $L_{\text{Cont}}$ term is omitted from $L_{\text{total}}$. The initial velocity and density are

$$\rho_0(x, y) = (z_1 + z_3)^2 + 1 \qquad v_0(x, y) = [e^{z_3}, e^{z_1}/2]$$

where $z = \theta(x, y, 1)$. Here we used the matrix-field parameterization as referred to in Section 2, for the simple reason that it performed better empirically. We also include the harmonic component from Hodge decomposition, which is just a constant vector field, to ensure we have full universality. Full methodology for the experiment is detailed in Appendix C. The results of evolving the flow for time $t \in [0, 1/3]$ are shown in Figure 4, as well as a comparison with a reference finite element solution generated by the conservative splitting method from [Guermond and Quartapelle, 2000]. In general, our model fits the initial conditions well and results in fluid-like movement of the density, however small approximation errors can accumulate from simulating over a longer time horizon. To highlight this, we also plotted the best result we were able to achieve with a standard PINN. Although the PINN made progress minimizing the loss, it was unable to properly model the dynamics of the fluid evolution. While this failure can likely be corrected by tuning the $\gamma$ constant (see Section 5.1), we were unable to find a combination that fit the equation correctly. By contrast, our NCL model worked with the first configuration we tried.

## 6 Learning the Hodge decomposition

Using our divergence-free models, we can also learn the Hodge decomposition of a vector field itself. Informally, the Hodge decomposition states that any vector field can be expressed as the combination

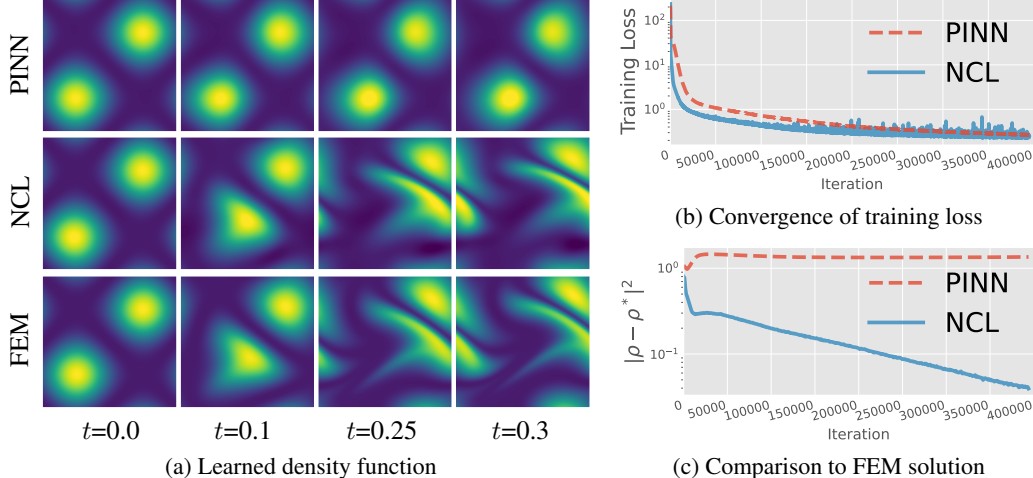

(a) Learned density function

(b) Convergence of training loss

(c) Comparison to FEM solution

Figure 4: While both PINN and NCL models minimize the training loss effectively and fit the initial conditions, the PINN fails to learn the dynamics of the advected density. When compared to a gold standard FEM solution, our NCL model nicely exhibits linear convergence to the solution.

of a gradient field (*i.e.* gradient of a scalar function) and a divergence-free field. There are many scenarios where a neural network modelling a vector field should be a gradient field (theoretically), such as score matching [Song and Ermon, 2019], Regularization by Denoising [Hurault et al., 2022], amortized optimization [Xue et al., 2020], and meta-learned optimization [Andrychowicz et al., 2016]. Many works, however, have reported that parameterizing such a field as the gradient of a potential hinders performance, and as such they opt to use an unconstrained vector field. Learning the Hodge decomposition can then be used to remove any extra divergence-free, *i.e.* rotational, components of the unconstrained model.

As a demonstration of the ability of our model to scale in dimension, we construct a synthetic experiment where we have access to the ground truth Hodge decomposition. This provides a benchmark to validate the accuracy of our method. To do so, we construct the target field $\hat{v}$ as

$$\hat{v} = \nabla w + \eta \tag{13}$$

where $\eta$ is a fixed divergence-free neural network, and $w$ is a fixed scalar network. We use the same embedding from Section 5.3, but extended to $\mathbb{T}^n$, for $n \in \{25, 50, 100\}$. We then parameterize another divergence-free model $v_\theta$ which has a different architecture than the target $\eta$. Inspired by the Hodge decomposition, we propose using the following objective for training:

$$\ell(\theta) = \|J_{\hat{v}-v_\theta} - [J_{\hat{v}-v_\theta}]^\mathsf{T}\|_F \tag{14}$$

where $J_{\hat{v}-v_\theta}$ denotes the Jacobian of $\hat{v} - v_\theta$. In the notation of differential forms, this corresponds to minimizing $\|\delta(\hat{v} - v_\theta)\|_F$, which when zero implies $v_\theta = \hat{v} + \delta\mu$, where $\mu$ is an arbitrary $n$-form. However, since $v_\theta$ is divergence-free, it must hold $\delta\mu = -\nabla w$, *i.e.* the left-over portion of $\hat{v} - v_\theta$ is exactly $\nabla w$, because the Hodge decomposition is an orthogonal direct sum [Warner, 1983].

For evaluation, we plot the squared $L_2$ norm to the ground truth $\eta$ against wall clock time. While this demonstration is admittedly artificial, it serves to show that our model scales efficiently to moderately high dimensions – significantly beyond those achievable using grid based discretizations of divergence-free fields (see Bhatia et al. [2013] for an overview of such methods). The results are shown in Figure 5. While larger dimensions are more costly, we are still able to recover the ground truth in dimensions up to 100.

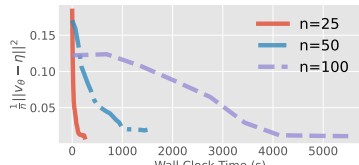

Figure 5: $L_2$ to ground truth.

# 7 Dynamical optimal transport

The calculation of optimal transport (OT) maps between two densities of interest $p_0$ and $p_1$ serves as a powerful tool for computing distances between distributions. Moreover, the inclusion of OT into machine learning has inspired multiple works on building efficient or stable generative models [Arjovsky et al., 2017, Tong et al., 2020, Makkuva et al., 2020, Huang et al., 2020, Finlay et al., 2020, Rout et al., 2021, Rozen et al., 2021]. In particular, we are interested in the dynamical formulation of Benamou and Brenier [2000], which is a specific instance where the transport map is characterized through a vector field. Specifically, we wish to solve the optimization problem

$$\min_{\rho, u} \int_0^1 \int_{\mathcal{M}} \|u(t, x)\|^2 \rho(t, x) \, dx dt \tag{15}$$

subject to the terminal constraints, $\rho(0, \cdot) = p_0(\cdot)$ and $\rho(1, \cdot) = p_1(\cdot)$, as well as remaining non-negative $\rho \geq 0$ on the domain and satisfying the continuity equation $\frac{\partial \rho}{\partial t} + \operatorname{div}(\rho u) = 0$.

In many existing cases, the models for estimating either $\rho$ and equation 15 are completely decoupled from the generative model or transport map [Arjovsky et al., 2017, Makkuva et al., 2020], or in some cases $\rho$ would be computed through numerical simulation and equation 15 estimated through samples [Chen et al., 2018, Finlay et al., 2020, Tong et al., 2020]. These approaches may not be ideal as they require an auxiliary process to satisfy or approximate the continuity equation.

In contrast, our NCL approach allows us to have access to a pair of $\rho$ and $u$ that always satisfies the continuity equation, without the need for any numerical simulation. As such, we only need to match the terminal conditions, and will automatically have access to a vector field $u$ that transports mass from $\rho(0, \cdot)$ to $\rho(1, \cdot)$. In order to find an optimal vector field, we can then optimize $u$ with respect to the objective in equation 15.

## 7.1 Parameterizing non-negative densities with subharmonic functions

In this setting, it is important that our density is non-negative for the entire domain and within the time interval between 0 and 1. Instead of adding another loss function to satisfy this condition, we show how we can easily bake this condition into our model through the vector-field parameterization.

Recall for the vector-field parameterization of the continuity equation, we have a vector field $b_\theta(y)$ with $y = [t, x]$, for $t \in [0, 1]$, $x \in \mathbb{R}^n$. This results in the parameterization

$$\rho = \operatorname{div}\left(\frac{\partial b_1}{\partial x} - \frac{\partial b_{2:n+1}}{\partial t}^\top\right) \quad \text{for } i = 1, \dots, n+1. \tag{16}$$

We now make the adjustment that $\frac{\partial b_{2:n+1}}{\partial t} = 0$. This condition does not reduce the expressivity of the formulation, as $\rho$ can still represent any density function. Interestingly, this modification does imply that $\rho = \nabla^2 b_1$ with $\nabla^2$ being the Laplacian operator, so that satisfying the boundary conditions is equivalent to solving a time-dependent Poisson's equation $\nabla^2 b_1(t, \cdot) = p_t$.

Furthermore, the class of *subharmonic* functions is exactly the set of functions $g$ that satisfies $\nabla^2 g(0, \cdot) \geq 0$. In one dimension, this corresponds to the class of convex functions but is a strict generalization of convex functions in higher dimensions. With this in mind, we propose using the parameterization

$$b_1(t, x) = \sum_{k=1}^{K} w_k(t) \phi(a_k(t) x + b_k(t)) \quad \text{where } \phi(z) = \frac{1}{4\pi}\left(\log(\|z\|^2) + E_1(\|z\|^2/4)\right) \tag{17}$$

with $E_1$ being the exponential integral function, and $w_k, a_k \in \mathbb{R}^+$ and $b_k \in \mathbb{R}^n$ are free functions of $t$. This a weighted sum of generalized linear models, where the choice of nonlinear function $\phi$ is chosen because it is the exact solution to the Poisson's equation for a standard normal distribution when $n = 2$, *i.e.* $\nabla^2 \phi = \mathcal{N}(0, I)$, while for $n \geq 2$ this remains a subharmonic function. As such, this can be seen as a generalization of a mixture model, which are often referenced to be universal density approximators, albeit requiring many mixture components [Goodfellow et al., 2016].

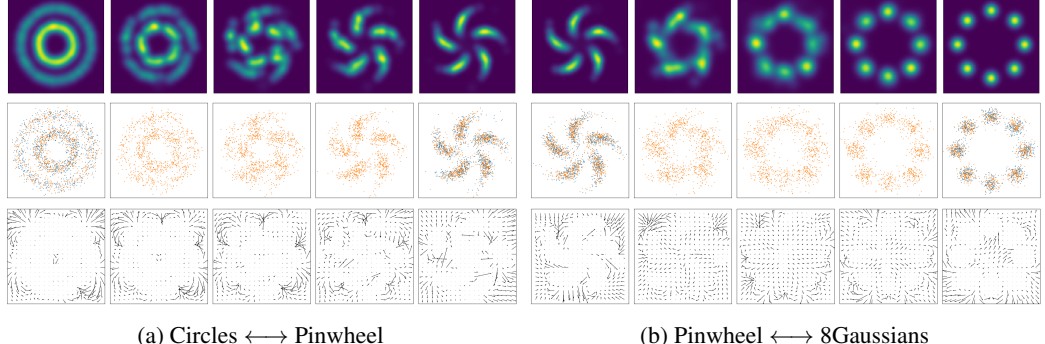

(a) Circles ⟷ Pinwheel                    (b) Pinwheel ⟷ 8Gaussians

Figure 6: Learned approximation to the dynamical optimal transport map. (*top*) Density $\rho$. (*mid*) Transformed samples in orange, with samples from $p_0$ and $p_1$ in blue. (*bottom*) Learned vector field.

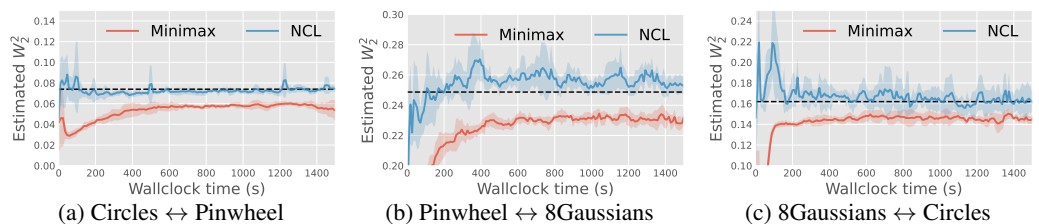

(a) Circles ↔ Pinwheel          (b) Pinwheel ↔ 8Gaussians          (c) 8Gaussians ↔ Circles

Figure 7: Neural Conservation Laws on approximating dynamical optimal transport converges in minutes. The alternative minimax formulation tends to underestimate the Wasserstein distance. Dashed line is an estimate from a discrete OT algorithm. Shaded regions denote standard deviation.

## 7.2 Experiments

We experiment with pairs of 2D densities (visualized in Figure 6) and fit a pair of $(\rho, u)$ to equation 15 while satisfying boundary conditions. Specifically, we train with the loss

$$\min_{\rho, u} \ \lambda \mathbb{E}_{x \sim \tilde{p}_0} \left[ |\rho(0, x) - p_0(x)| \right] + \lambda \mathbb{E}_{x \sim \tilde{p}_1} \left[ |\rho(1, x) - p_1(x)| \right] + \int_0^1 \int_{\mathcal{M}} \|u(t, x)\|^2 \rho(t, x) \ dx dt \tag{18}$$

where $\tilde{p}_i$ is a mixture between $p_i$ and a uniform density over a sufficiently large area, for $i = 0, 1$, and $\lambda$ is a hyperparameter. We use a batch size of 256 and set $K$=128 (from equation 17). Qualitative results are displayed in Figure 6, which include interpolations between pairs of densities.

Furthermore, we can use the trained models to estimate the Wasserstein distance, by transforming samples from $p_0$ through the learned vector field to time $t$=1. We do this for 5000 samples and compare the estimated value to (i) one estimated by that of a neural network trained through a minimax optimization formulation [Makkuva et al., 2020], and (ii) one estimated by that of a discrete OT solver [Bonneel et al., 2011] based on kernel density approximation and interfaced through the `pot` library [Flamary et al., 2021].

Estimated squared Wasserstein distance as a function of wallclock time are shown in Figure 7. We see that our estimated values roughly agree with the discrete OT algorithm. However, the baseline minimax approach consistently underestimates the optimal transport distance. This is an issue with the minimax formulation not being able to cover all of the modes, and has been commonly observed in the literature (*e.g.* Salimans et al. [2016], Che et al. [2017], Srivastava et al. [2017]). Moreover, in order to stabilize minimax optimization in practice, Makkuva et al. [2020] made use of careful optimization tuning such as reducing momentum [Radford et al., 2016]. In contrast, our NCL approach is a simple optimization problem.

# 8    Conclusion

We proposed two constructions for building divergence-free neural networks from an original unconstrained smooth neural network, where either an original matrix field or vector field is parameterized. We found both to be useful in practice: the matrix field approach is generally more flexible, while the vector field approach allows the addition of extra constraints such as a non-negativity constraint. We combined these methods with the insight that the continuity equation can reformulated as a divergence-free vector field, and this resulted in a method for building deep neural networks that are constrained to always satisfy the continuity equation, which we refer to as Neural Conservation Laws.

Currently, these models are difficult to scale due to extensive use of automatic differentiation for computing divergence of neural networks. It may be possible to combine our approach with parameterizations that provide cheap divergence computations [Chen and Duvenaud, 2019]. Nevertheless, the development of efficient batching and automatic differentiation tools is an active research area, so we expect these tools to improve as development progresses.

## Acknowledgements

We acknowledge the Python community [Van Rossum and Drake Jr, 1995, Oliphant, 2007] and the core set of tools that enabled this work, including PyTorch [Paszke et al., 2019], functorch [Horace He, 2021], torchdiffeq [Chen, 2018], JAX [Bradbury et al., 2018], Flax [Heek et al., 2020], Hydra [Yadan, 2019], Jupyter [Kluyver et al., 2016], Matplotlib [Hunter, 2007], numpy [Oliphant, 2006, Van Der Walt et al., 2011], and SciPy [Jones et al., 2014]. Jack Richter-Powell would also like to thank David Duvenaud and the Vector Institute for graciously supporting them over the last year.

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
