# A Preliminaries: Differential forms in $\mathbb{R}^n$

We provide an in-depth explanation of our divergence-free construction in this section.

We will use the notation of *differential forms* in $\mathbb{R}^n$ that will help derivations and proofs in the paper. Below we provide basic definitions and properties of differential forms, for more extensive introduction see *e.g.*, [Do Carmo, 1998, Morita, 2001].

We let $x = (x_1, \ldots, x_n) \in \mathbb{R}^n$, and $dx_1, \ldots, dx_n$ the coordinate differentials, *i.e.*, $dx_i(x) = x_i$ for all $i \in [n] = \{1, \ldots, n\}$. The linear vector space of $k$-forms in $\mathbb{R}^n$, denoted $\Lambda^k(\mathbb{R}^n)$, is the space of $k$-linear alternating maps

$$\varphi : \overbrace{\mathbb{R}^n \times \cdots \times \mathbb{R}^n}^{k \text{ times}} \to \mathbb{R} \tag{19}$$

A $k$-linear alternating map $\varphi$ is linear in each of its coordinates and satisfies $\varphi(\ldots, v, \ldots, u, \ldots) = -\varphi(\ldots, u, \ldots, v, \ldots)$. The space $\Lambda^k(\mathbb{R}^n)$ is a linear vector space with a basis of alternating $k$-forms denoted $dx_{i_1} \wedge \cdots \wedge dx_{i_k}$. The way these $k$-forms act on $k$-vectors, $v_1, \ldots, v_k \in \mathbb{R}^n$ is as a signed volume function:

$$dx_{i_1} \wedge \cdots \wedge dx_{i_k}(v_1, \ldots, v_k) = \det \left[ dx_{i_r}(v_s) \right]_{r,s \in [k]} \tag{20}$$

Expanding an arbitrary element $\omega \in \Lambda^k(\mathbb{R}^n)$ in this basis gives

$$\omega = \sum_{i_1 < i_2 < \cdots < i_k} a_{i_1 \cdots i_k} \, dx_{i_1} \wedge \cdots \wedge dx_{i_k} = \sum_I a_I \, dx_I \tag{21}$$

where $i_1, \ldots, i_k \in [n]$, $I = (i_1, \ldots, i_k)$ are multi-indices, $a_I$ are scalars, and $dx_I = dx_{i_1} \wedge \cdots \wedge dx_{i_k}$.

The space of *differential $k$-forms* (also called $k$-forms in short), denoted $\mathcal{A}^k(\mathbb{R}^n)$, is defined by smoothly assigning to each $x \in \mathbb{R}^n$ a $k$-linear alternating form $w \in \Lambda^k(\mathbb{R}^n)$. That is

$$w(x) = \sum_I f_I(x) dx_I \tag{22}$$

where $f_I : \mathbb{R}^n \to \mathbb{R}$ are smooth scalar functions. Note that $\mathcal{A}^0(\mathbb{R}^n)$ is the space of smooth scalar functions over $\mathbb{R}^n$. The differential operator can be seen as a linear operator $d : \mathcal{A}^0(\mathbb{R}^n) \to \mathcal{A}^1(\mathbb{R}^n)$ defined by

$$df(x) = \sum_{i=1}^n \frac{\partial f}{\partial x_i}(x) dx_i \tag{23}$$

The exterior derivative $d : \mathcal{A}^k(\mathbb{R}^n) \to \mathcal{A}^{k+1}(\mathbb{R}^n)$ is a linear differential operator generalizing the differential to arbitrary differential $k$-forms:

$$d\omega(x) = \sum_I df_I \wedge dx_I \tag{24}$$

where the exterior product $\omega \wedge \eta$ of two forms $\omega = \sum_I f_I dx_I$, $\eta = \sum_J g_J dx_J$ is defined by extending equation 20 linearly, that is, $\omega \wedge \eta = \sum_{I,J} f_I g_J dx_I \wedge dx_J$. An imporant property of the exterior derivative is that $dd\omega = 0$ for all $\omega$. This property can be checked using the definition in equation 24 and the symmetry of mixed partials, $\frac{\partial^2 f}{\partial x_i \partial x_j} = \frac{\partial^2 f}{\partial x_j \partial x_i}$.

The hodge operator $\star : \mathcal{A}^k(\mathbb{R}^n) \to \mathcal{A}^{n-k}(\mathbb{R}^n)$ matches to each $k$-form an $n - k$-form by extending the rule

$$\star (dx_I) = (-1)^\sigma dx_J \tag{25}$$

linearly, where $\sigma = \text{sign}([I, J])$ is the sign of the permutation $(I, J) = (i_1, \ldots, i_k, j_1, \ldots, i_{n-k})$ of $[n]$. The hodge star is (up to a sign) its own inverse, $\star \star \omega = (-1)^{k(n-k)} \omega$, for all $\omega \in \mathcal{A}^k(\mathbb{R}^n)$. The $d$ operator has an adjoint operator $\delta : \mathcal{A}^k(\mathbb{R}^n) \to \mathcal{A}^{k-1}(\mathbb{R}^n)$ defined by

$$\delta = (-1)^{n(k+1)+1} \star d \star . \tag{26}$$

A vector field $v(x) = (v_1(x), \ldots, v_n(x))$ in $\mathbb{R}^n$ can be identified with a 1-form $v = \sum_{i=1}^n v_i(x) dx_i$. The divergence of $v$, denoted $\text{div}(v)$ can be expressed using exterior derivative :

$$d \star v = d \sum_{i=1}^n v_i \star dx_i = \sum_{i=1}^n \frac{\partial v_i}{\partial x_i} dx_1 \wedge \cdots \wedge dx_n = \text{div}(v) dx_1 \wedge \cdots \wedge dx_n, \tag{27}$$

where in the second equality we used the definition of $\star$, the definition of $d$ and the fact that $dx_i \wedge dx_i = 0$ (can be seen from the repeating rows in the matrix inside the determinant in equation 20). Note that the $n$-form $\mathrm{div}(v)dx_1 \wedge \cdots \wedge dx_n$ is identified via $\star$ with the function (*i.e.*, 0-form) $\mathrm{div}(v)$.

### A.1 Derivations of equation 3 and equation 6

We will need the following identity below, for $\mu \in \Omega^k(\mathcal{M})$, expressed as

$$\mu = \sum_I \alpha_I \star (dx^I)$$

we have

$$\star \mu = (-1)^{k(n\text{-}k)} \sum_I \alpha_I dx^I \tag{28}$$

which follows by linearity of $\star$ over $C^\infty(\mathcal{M})$ and the identity $\star\star = (-1)^{k(n-k)}\mathrm{id}$.

Here, we derive the expression for equation 3. For $\mu \in \Omega^{n-2}(\mathcal{M})$, the exterior derivative is given by equation 24. However, in this case, since $dx^i \wedge dx^i = 0$, we can expand

$$d\mu = \sum_{i,j} d\mu_{i,j} \wedge \star(dx^i \wedge dx^j) \tag{29}$$

$$= \sum_{i,j} \sum_k \frac{\partial \mu_k}{\partial x^k} dx^k \wedge \star(dx^i \wedge dx^j) \tag{30}$$

$$= \sum_{i,j} \left[ \frac{\partial \mu_{ij}}{\partial x^i} dx^i \wedge \star(dx^i \wedge dx^j) + \frac{\partial \mu_{ij}}{\partial x^j} dx^j \wedge \star(dx^i \wedge dx^j) \right] \tag{31}$$

$$= \sum_{i,j} \left[ \frac{\partial \mu_{ij}}{\partial x^i} dx^i \wedge \star(dx^i \wedge dx^j) \right] + \sum_{i,j} \left[ \frac{\partial \mu_{ji}}{\partial x^i} dx^i \wedge \star(dx^j \wedge dx^i) \right] \tag{32}$$

$$= \sum_{i,j} \left[ \frac{\partial \mu_{ij}}{\partial x^i} dx^i \wedge \star(dx^i \wedge dx^j) \right] + \sum_{i,j} \left[ \frac{\partial \mu_{ij}}{\partial x^i} dx^i \wedge \star(dx^i \wedge dx^j) \right] \tag{33}$$

$$= 2 \sum_{i,j} \left[ \frac{\partial \mu_{ij}}{\partial x^i} dx^i \wedge \star(dx^i \wedge dx^j) \right] \tag{34}$$

$$= 2 \sum_i \left[ \sum_j \frac{\partial \mu_{ij}}{\partial x^j} \star (dx^i) \right] \tag{35}$$

where equation 32 follows by re-indexing $i$ as $j$, and then equation 33 by anti-symmetry $\mu_{ij} = -\mu_{ji}$ and $\star(dx^i \wedge dx^j) = -\star(dx^j \wedge dx^i)$ (the sign flips cancel). Applying $\star$ then gives (by equation 28),

$$\star d\mu = \star \left[ 2 \sum_i \sum_j \frac{\partial \mu_{ij}}{\partial x^j} \star (dx^i) \right] = 2(-1)^{n-1} \sum_i \sum_j \frac{\partial \mu_{ij}}{\partial x^j} dx^i$$

To derive equation equation 6, it suffices to start by noting $\delta = \star d\star$, and so

$$\delta\nu = \star d \star \nu \tag{36}$$

$$= \star d \star \sum_i \nu_i \star (dx^i) \tag{37}$$

$$= \star d(-1)^{n-1} \sum_i \nu_i dx^i \tag{38}$$

$$= (-1)^{n-1} \star \sum_i \sum_j \frac{\partial \nu_i}{\partial x^j} dx^j \wedge dx^i \tag{39}$$

$$= (-1)^{n-1} \sum_i \sum_j \frac{\partial \nu_i}{\partial x^j} \star (dx^j \wedge dx^i) \tag{40}$$

$$= -(-1)^{n-1} \sum_i \sum_j \frac{\partial \nu_i}{\partial x^j} \star (dx^i \wedge dx^j) \tag{41}$$

$$= -(-1)^{n-1} \left( \sum_{i<j} \frac{\partial \nu_i}{\partial x^j} \star (dx^i \wedge dx^j) + \sum_{i>j} \frac{\partial \nu_i}{\partial x^j} \star (dx^i \wedge dx^j) \right) \tag{42}$$

$$= -(-1)^{n-1} \left( \sum_{i<j} \frac{\partial \nu_i}{\partial x^j} \star (dx^i \wedge dx^j) - \sum_{i<j} \frac{\partial \nu_j}{\partial x^i} \star (dx^i \wedge dx^j) \right) \tag{43}$$

$$= -(-1)^{n-1} \sum_{i<j} \left[ \frac{\partial \nu_i}{\partial x^j} - \frac{\partial \nu_j}{\partial x^i} \right] \star (dx^i \wedge dx^j) \tag{44}$$

then since $\star(dx^i \wedge dx^j) = -\star(dx^j \wedge dx^i)$, we have

$$-(-1)^{n-1} \sum_{i<j} \left[ \frac{\partial \nu_i}{\partial x^j} - \frac{\partial \nu_j}{\partial x^i} \right] \star (dx^i \wedge dx^j) \tag{45}$$

$$= -(-1)^{n-1} \left( \frac{1}{2} \sum_{i<j} \left[ \frac{\partial \nu_i}{\partial x^j} - \frac{\partial \nu_j}{\partial x^i} \right] \star (dx^i \wedge dx^j) + \frac{1}{2} \sum_{i<j} \left[ \frac{\partial \nu_j}{\partial x^i} - \frac{\partial \nu_i}{\partial x^j} \right] \star (dx^j \wedge dx^i) \right) \tag{46}$$

$$= -(-1)^{n-1} \left( \frac{1}{2} \sum_{i<j} \left[ \frac{\partial \nu_i}{\partial x^j} - \frac{\partial \nu_j}{\partial x^i} \right] \star (dx^i \wedge dx^j) + \frac{1}{2} \sum_{i>j} \left[ \frac{\partial \nu_i}{\partial x^j} - \frac{\partial \nu_j}{\partial x^i} \right] \star (dx^i \wedge dx^j) \right) \tag{47}$$

$$= -(-1)^{n-1} \left( \frac{1}{2} \sum_{i,j} \left[ \frac{\partial \nu_i}{\partial x^j} - \frac{\partial \nu_j}{\partial x^i} \right] \star (dx^i \wedge dx^j) \right) \tag{48}$$

$$\tag{49}$$

# B  Proofs and derivations

## B.1  Universality

**Theorem 2.1.** *The matrix and vector-field representations are universal in $\mathbb{T}$, possibly only missing a constant vector field.*

*Proof.* First, consider a divergence-free vector field and its representation as a 1-form $v \in \mathcal{A}^1(\mathbb{T})$. Then $v$ being divergence-free means $d \star v = 0$. We denote by $c = (c_1, \dots, c_n) \in \mathbb{R}^n$ a constant vector field; note that $c$ is also a well defined vector field over $\mathbb{T}$. We will use the notation $c$ to denote the corresponding constant 1-form. We claim $\star v = d\mu + \star c$, where $\mu \in \mathcal{A}^{n-2}(\mathbb{T})$, and $\star c \in \mathcal{A}^{n-1}(\mathcal{S})$

is constant. This can be shown with Hodge decomposition [Morita, 2001] of $\star v \in \mathcal{A}^{n-1}(\mathbb{T})$:

$$\star v = d\mu + \delta\tau + h, \tag{50}$$

where $\mu \in \mathcal{A}^{n-2}(\mathbb{T})$, $\tau \in \mathcal{A}^n(\mathbb{T})$ and $h \in \mathcal{A}^{n-1}$ is a harmonic $n$-1-form. Note that the harmonic $n$-1-forms over $\mathcal{T}$ are simply constant. Taking $d$ of both sides leads to

$$d \star v = d\delta\tau. \tag{51}$$

Since we assumed $v$ is div-free, $d \star v = 0$ and we get $d\delta\tau = 0$. Since $d\delta\tau = 0 = \delta\delta\tau$ then $\delta\tau$ is harmonic (constant) as well.

So far we showed $\star v = d\mu + \star c$, which shows the universality of the matrix-field representation (up to a constant). To show universality of the vector-field representation we need to show that:

$$\left\{ d\mu + \star c \mid \mu \in \mathcal{A}^{n-2}(\mathbb{T}) \right\} = \left\{ d\delta\nu + \star c \mid \nu \in \mathcal{A}^{n-1}(\mathbb{T}) \right\} \tag{52}$$

The left inclusion $\supset$ is true since $\delta\nu \in \mathcal{A}^{n-2}(\mathcal{T})$. For the right inclusion $\subset$ we take an arbitrary $\mu \in \mathcal{A}^{n-2}(\mathbb{T})$ and decompose it with Hodge: $\mu = d\omega + \delta\nu + h$, where $\omega \in \mathcal{A}^{n-3}(\mathbb{T})$, $\nu \in \mathcal{A}^{n-1}(\mathbb{T})$, and $h \in \mathcal{A}^{n-2}(\mathbb{T})$ is harmonic. Taking $d$ of both sides leaves us with $d\mu = d\delta\nu$ that shows that $d\mu + \star c$ is included in the right set.

$\square$

## B.2 Stabilizing training for fluid simulations

In order to stabilize training, we can modify the loss terms $L_F, L_G, L_I$ to avoid division by $\rho$. As before $v = [\rho, \rho u]$,and

$$\tilde{L}_F = \left\| \rho^2 (\rho u)_t - \rho(\rho_t)\rho u + \rho[D(\rho u)(\rho u)] - [\nabla\rho \otimes \rho u](\rho u) + \rho^2 \nabla p \right\|_\Omega^2 \tag{53}$$

$$\tilde{L}_{\text{div}} = \left\| \tilde{\nabla}\rho \cdot v \right\|_\Omega \tag{54}$$

$$\tilde{L}_I = \| \rho u(0, \cdot) - \rho_0 u_0(0, \cdot) \|_\Omega^2 + \| \rho(0, \cdot) - \rho_0(0, \cdot) \|_\Omega^2 \tag{55}$$

$$\tilde{L}_G = \| \rho u \cdot n \|_{\partial\Omega}^2 \tag{56}$$

In practice, we noticed this improved training stability significantly, which is intuitive since the possibility of a division by 0 is removed. The derivation of $\tilde{L}_G$, and $\tilde{L}_I$ is simply scaling by $\rho$. We derive $\tilde{L}_F$ and $\tilde{L}_{\text{div}}$ by repeatedly applying the product rules for the Jacobian and divergence operators and solving for $\rho^{2,3}$ scaled copies of the residuals. Below, we use the convention that the gradient and divergence operators only act in spatial variables, but the $\tilde{\nabla}$, $\tilde{\text{div}}$ operators include time

$$\tilde{\nabla}\rho(t, x) = \begin{pmatrix} \frac{\partial\rho}{\partial t}(t, x) \\ \frac{\partial\rho}{\partial x_1}(t, x) \\ \vdots \end{pmatrix} \qquad \tilde{\text{div}}(u) = \frac{\partial u_0}{\partial t} + \sum_{i=1}^{d} \frac{\partial u_i}{\partial x_i} \tag{57}$$

**Derivation of $\tilde{L}_{\text{div}}$:**
To derive $\tilde{L}_{\text{div}}$, we consider

$$\tilde{\text{div}}\left( \rho \begin{pmatrix} 1 \\ u \end{pmatrix} \right) = \tilde{\nabla}\rho \cdot \begin{pmatrix} 1 \\ u \end{pmatrix} + \rho\tilde{\text{div}}\begin{pmatrix} 1 \\ u \end{pmatrix} \tag{58}$$

by construction $\text{div}\left( \rho \begin{pmatrix} 1 \\ u \end{pmatrix} \right) = 0$, and since $\tilde{\text{div}}\begin{pmatrix} 1 \\ u \end{pmatrix} = \text{div}(u)$, multiplying both sides by $\rho$ we find

$$0 = \tilde{\nabla}\rho \cdot v + \rho^2 \text{div}(u) \implies \rho^2 \text{div}(u) = -\tilde{\nabla}\rho \cdot v \tag{59}$$

**Derivation of $\tilde{L}_F$** To derive $\tilde{L}_F$, we will start at the end. We multiply the momentum term of the Euler system

$$\frac{\partial u}{\partial t} + [Du]u + \frac{\nabla p}{\rho} = 0 \implies \rho^3 \frac{\partial u}{\partial t} + \rho^3[Du]u + \rho^2\nabla p = 0 \tag{60}$$

we start by applying the product rule to $\rho u$

$$\frac{\partial(\rho u)}{\partial t} = \frac{\partial \rho}{\partial t} u + \rho \frac{\partial u}{\partial t} \tag{61}$$

multiplying by $\rho^2$ and solving for $\rho^3 \frac{\partial u}{\partial t}$ yields

$$\rho^3 \frac{\partial u}{\partial t} = \rho^2 \frac{\partial(\rho u)}{\partial t} - \rho \frac{\partial \rho}{\partial t} \rho u \tag{62}$$

which can be computed without dividing by $\rho$. Now we apply the Jacobian scalar product rule

$$D(\rho u) = \nabla \rho \otimes u + \rho Du \tag{63}$$

contracting with $\rho^2 u$ yields that

$$D(\rho u)\rho^2 u = [\nabla p \otimes (\rho u)](\rho u) + \rho^3 [Du]u \tag{64}$$

which gives

$$\rho^3 [Du]u = D(\rho u)\rho^2 u - [\nabla p \otimes (\rho u)](\rho u) \tag{65}$$

which can also be computed without invoking division by $\rho$. Together, equation 62 and equation 65 yield $\tilde{L}_F$.

## C   Implementation Details

### C.1   Tori Example Details

For the Tori example, we used the matrix formulation of the NCL model. The matrix was parameterized as the output of a 8 layer, 512-wide Multi-Layer Perceptron with softplus activation. We trained this model for 600,000 steps of stochastic gradient descent using a batch size of 1000. The weight vector $\gamma$ was fixed with $\gamma_F = 3 \times 10^{-3}$, $\gamma_I = 30$, $\gamma_{\text{div}} = 0.01$. Instead of choosing a fixed set of colocation points (as is common in the PINN literature see Raissi et al. [2017]), we sampled uniformly on the unit square $[0, 1]^2$. For the Finite-Element reference solution, we solved the system on a 50 x 50 grid with periodic boundary conditions implemented with a mixed Lagrange element scheme. The splitting scheme used was the inviscid case of Guermond and Quartapelle [2000], with time step $dt = 0.001$.

### C.2   3d Ball Example Details

For the comparison in Figure 9 we used a 4-layer, 128 wide feed forward network for both the Curl PINN and the NCL. Stopped training both models after 10000 steps of stochastic gradient descent with batch size of 1000. While this is much less than Section 5.3, the difference can be explained by the initial condition being much less complex. For the NCL model, we used $\gamma_F = 0.1, \gamma_{\text{div}} = 0.1, \gamma_G = 0.1, \gamma_I = 30$. For the CURL model, we used $\gamma_F = 0.1, \gamma_G = 0.1, \gamma_{\text{cont}} = 10, \gamma_I = 30$

A larger plot of the comparison is shown in Figure 9.

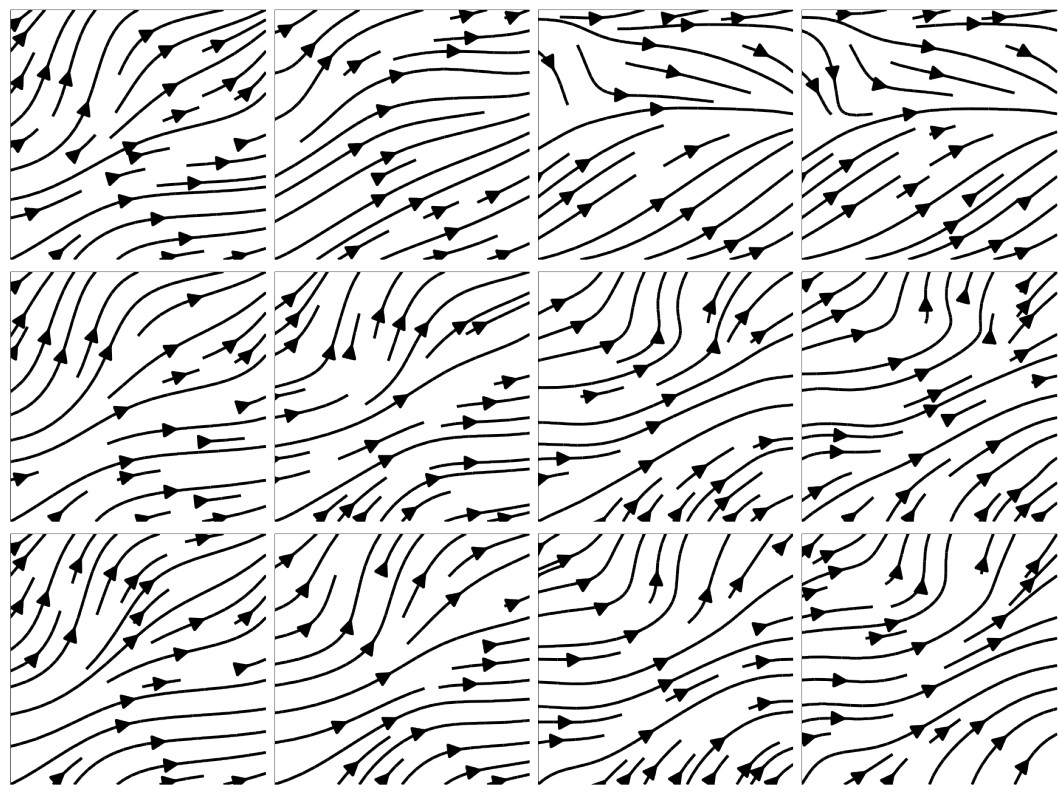

Figure 8: Streamplots showing the velocity field from a Physics Informed Neural Network (*top*), our method (*middle*) vs a reference FEM solution (*bottom*). While both models minimize the loss effectively and fit the initial conditions, the PINN fails to learn the correct evolution of the velocity.

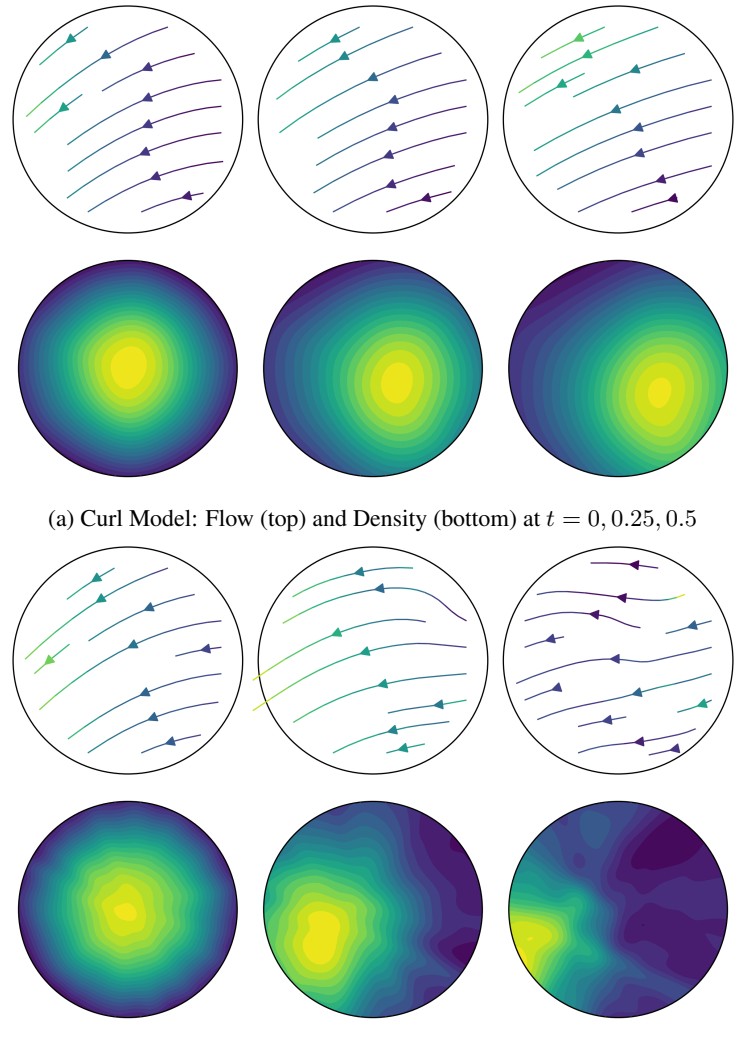

(a) Curl Model: Flow (top) and Density (bottom) at $t = 0, 0.25, 0.5$

(b) Our model (NCL): Flow (top) and Density (bottom) at $t = 0, 0.25, 0.5$

Figure 9: Larger version of comparison shown in Section 5.2. While our model fits the initial conditions and convects the density along the flow lines as expected, the curl model fails to do so.

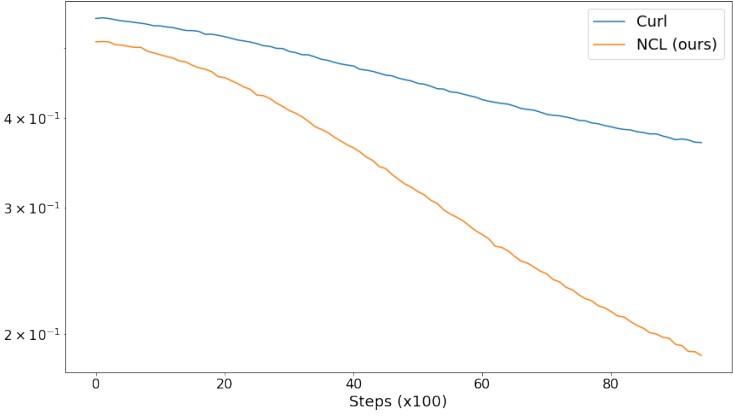

Figure 10: Training loss for NCL model (ours) plotted against the Curl model, for the 3D unit ball fluid experiment. Both models achieve a similar order of magnitude loss but exhibit qualitatively different results.