# OpenReview forum: "Neural Conservation Laws: A Divergence-Free Perspective"
_NeurIPS.cc/2022/Conference — NeurIPS 2022 Accept_

### Official Review · Reviewer_xPeb · 2022-06-23

**Rating:** 7
**Confidence:** 3
**Soundness:** 3 good
**Presentation:** 2 fair
**Contribution:** 4 excellent

**Summary:**

The paper introduces two parametrizations for Divergence-Free Vector Fields in arbitrary dimensions. In principle, this offers an extension to the well-known method of using the curl of a vector field as a divergence-free vector field in $\mathbb{R}^3$.
The derivation of such Divergence-Free vector fields is based on Differential Forms and seemingly follows from first principles within this area.
The two different parametrizations were similarly derived but come with different properties.
a) is based on a Matrix representation, which scales quadratically with the input dimension, and b) requires 2nd order derivatives but only relies on a more compact vector representation, and further allows "building in" constraints such as positivity.

Significant applications for such a parameterization include the modeling of compressible fluids and the Maxwell Equations.
The velocity $\vec{v}$ and density $\rho$ of a mass-conserving fluid can be reformulated in terms of a divergence-free vector-field $\vec{v}' = [\rho, \rho \cdot \vec{v}]$ with $\vec{v}\in\mathbb{R}^3$ using the augmented arguments $[t, x, y, z]$.
Consequently, the proposed parameterization enables building a Neural Network, which by construction conserves mass instead of relying on soft constraints enforced similar to Physics Informed Neural Networks (PINNs).
A similar construction can be made for e.g. the Maxwell equation.

As a second application setting, the modeling of probability densities in an optimal transport setting is showcased.
For this, one of the introduced parameterizations is adapted to conform to the density constraints by construction (positivity, within 0 and 1).

Mostly qualitative experiments in simple settings are shown for both settings, providing a proof of concept for modeling Euler equations on a Ball and Torus, and learning optimal transport maps.
For the Euler equations, improved modeling of the flow compared to a baseline PINN is shown.


**Questions:**

- Considering fluid flows in 2D, the augmented vector field would be in 3D and your equation would correspond to the curl: $*d\mu = \text{curl}(\vec{v})$, with a 3D vector representation $\vec{v}\in\mathbb{R}^3$ requiring only first-order gradients. How would this compare to the proposed matrix- and vector-field representations in praxis? Would the matrix representation still be favorable, as it is again more flexible in low dimensions?
This would be interesting since many real applications consider a simplified problem flattened to 2-dimensional domains.



**Limitations:**

- The authors mention possible scaling limitations of the individual proposed parametrization, namely the required number of entries for the matrix representations and the higher-order derivatives needed for the vector representation.

- Possible stability issues due to the division of $\rho$ for some losses are indirectly addressed by reformulating the loss terms. When access to the velocity is directly needed (e.g. observations of the velocity are provided for the training), training instabilities might still be an issue.

- It is not totally clear how the performance and flexibility of the two proposed parametrizations differ in praxis or how they behave during training. The authors provide a vague statement that the matrix representation is more flexible in lower-dimensional settings. The vector field representation offers some options for adding additional constraints such as positivity.

**Strengths And Weaknesses:**

### Significance:
1. _**[Medium]**_ A general parametrization for Divergence-Free Vector Fields in arbitrary Dimensions is proposed.
2.  _**[Medium]**_ The authors introduce to the ML community an elegant reformulation of continuity equations, that allows reformulating the problem to finding divergence-free vector fields.
3.  _**[High]**_ Combining (1.) and (2.), a Neural Network Architecture that provides conservation of mass by construction is provided. Enforcing this conservation constraint in Physics Informed Neural Networks (PINNs) is a difficult problem, limiting in my experience the number of publications of PINNs made within the area of compressible fluid dynamics. This work may open up PINNs to difficult problem settings such as Compressible Navier Stokes equations.
4.  _**[High]**_ Combining (1.) and (2.), allows for solving optimal transport optimization problems, while by construction enforcement of the mass conservation constraints.  Other methods often require additional enforcement of these constraints, with non-adherence leading to a meaningless optimization objective.

### Originality:
1. The proposed construction of Divergence-free Vector Fields (of arbitrary dimensionality) with Neural Networks is novel.
2. The insight of reformulating the conservation of mass equation $\frac{\delta \rho}{\delta t} + div( \rho \cdot \vec{v} )=0$ to $div(\vec{v_{aug}})=0$ is to the best of my knowledge novel to the Machine Learning community.

### Quality:
- To the best of my knowledge, the proposed approach is technically sound and complete.
- The experiments are sound and provide a qualitative proof of concept of the provided method.
- The universality proof provided in section 3.1 seems sound, however, I am rather limited in my familiarity with differential forms.
- No code is provided

### Clarity:
- The general story and flow of the paper is well organized and readable.
- The preliminaries on differential forms are not well written and structured.
Although the mathematical tools for the derivation of the divergence-free vector fields are provided, they are not referenced when used and intermediate steps are skipped.
Instead, they are handwaved in the form of "direct calculation shows[...]" (l. 96, l. 106).
When not that familiar with differential forms, it will be difficult to follow the paper in the current form - as it is the case for me.
Differential forms are in my opinion not a well-known area within ML, and such derivations should be made more clearly - A more step-wise derivation in the Appendix would help significantly.
- The notation is at times confusing, e.g. with overloaded variables (l.103), or missing equation numbers (e.g. between l.96 and l.97)
- (Small note on Terminology) "Incompressible Flow". A short clarification of what incompressible flows are and how they are not the same as incompressible fluids is probably helpful to avoid confusion in a wider audience.


To summarize, this is a very promising work that is in my opinion a significant step for applying Physics Informed Neural Networks to challenging compressible fluid dynamics problems.
While the contribution itself is of high quality, novelty and significance, I see issues in terms of the clarity of writing mainly in the provided background in differential forms.

---

> ### Author Response · Authors · 2022-08-02
> **Response to Reviewer xPeb**
>
> We thank the reviewer for their feedback and for giving a very thorough review of our work. We largely share the reviewer’s enthusiasm and thank them for their encouragement. We also agree with the reviewer that the presentation may not have been ideal (for the ML community), and have revised the submission (mostly rearranging sections and adding more clarifications) to hopefully increase our readability for the ML community.
>
> > The preliminaries on differential forms are not well written and structured… Differential forms are in my opinion not a well-known area within ML, and such derivations should be made more clearly - A more step-wise derivation in the Appendix would help significantly.
>
> We prepared this mainly for readers who have encountered differential forms before, in which case this section would help them brush up on the topic. We realize this section may not be useful for the majority of readers and have instead moved it to the Appendix. This allows us to streamline the narrative to directly discuss the construction approaches, while full proofs are left in the Appendix. We also plan on adding the step-wise derivations to the Appendix, but did not have time during the rebuttal. Overall, we think this has significantly improved the clarity of our paper and we thank the reviewer for bringing these issues up.
>
> > Limitations
>
> We have added further clarifications and a plot to discuss the scaling issues for these two constructions. The division by rho during training was remedied completely due to the loss reformulation. However, we note that dividing by rho is still required to extract the vector field, which can be problematic if we ever wish to simulate (and leads to a bit of error in the optimal transport experiment). Overall, we felt we were not hiding these limitations, and will aim to improve clarity on these issues (as they also lead into future directions).

---

> > ### Comment · Reviewer_xPeb · 2022-08-08
> > **Response**
> >
> > Thank you for the response and the revision of the paper.
> >
> > I think the decision for moving the background to differential forms to the appendix makes sense and provides more room for the experiments and the very general background. I do recommend including step-wise derivations in the final version if time allows it - it would make this paper much more accessible.

---

### Official Review · Reviewer_Mgw4 · 2022-07-12

**Rating:** 5
**Confidence:** 2
**Soundness:** 3 good
**Presentation:** 3 good
**Contribution:** 3 good

**Summary:**

This paper describes a method for parameterizing divergence-free vector fields in N-dimensions. The method is demonstrated with application to physics-informed neural networks (PINNs) for modeling the 2D Euler Equations with non-constant density, and to 2D optimal transport.

**Questions:**

1. Any there any applications you can think of that would need more than 4 dimensions?
2. Can you add quantitative comparisons between the matrix-field and vector-field formulations?

**Limitations:**

No concerns.

**Strengths And Weaknesses:**

My experience is more on the PDE side, so I'll focus my review there. (Hopefully other reviewers can cover the optimal transport example.)

Strengths:

1. The paper describes two general approaches to parameterizing N-D divergence free fields. This will be a good references for the physics/ML community.
2. The paper demonstrates that the approach is workable in several machine learning applications.

Weaknesses:

1. The proof is for N-dimensional divergence free fields, but the examples are all for 2+1 or 3+1 space+time dimensions. Exactly one example (the 3D ball) requires going beyond the very well approach of parameterizing a vector field and taking the curl. A lack of high-dimensional examples (or even hypothetical high-dimensional applications) makes the discussion about $O(n)$ vs $O(n^2)$ scaling with dimensionality seem a little silly.
2. There are no quantitative comparisons between the two different parameterizations of divergence-free fields (matrix-free and vector-free). This would strengthen the paper.
3. I am concerned about the applicability for PDE problems. The paper describes avoiding "expensive numerical simulations" by parameterizing solutions with a neural network as a strength, but in truth PINNs are orders of magnitude [slower and less accurate](https://arxiv.org/abs/2205.14249) than state of the art simulation methods that parameterize solutions on grids. PINNs have indeed received a fair amount of recent attention from the physics community, but are not yet really a practical method, e.g., as demonstrated by the poor results in this paper for modeling the 2D flow shown in Figure 2.
4. The application to the Euler equation with non-constant density is also rather niche. In my experience, it much more common to either enforce constant density, in which case the curl formulation for vector fields is sufficient (as in Rao 2020), or to solve the full compressible Navier-Stokes via an equation of state. The technique in this paper would be application for this later case, which would probably make more compelling examples.

---

> ### Author Response · Authors · 2022-08-02
> **Response to Reviewer Mgw4**
>
> We thank the reviewer for their feedback. We largely agree with the reviewer on concerns regarding scale and the weakness of PINN-based approaches. However, we argue that we’re *fixing* the unreliable nature of PINNs by directly enforcing PDEs as constraints. We have updated the paper to address these concerns.
>
> > The proof is for N-dimensional divergence free fields, but the examples are all for 2+1 or 3+1 space+time dimensions.
>
> We have added experiments on learning the Hodge decomposition for moderate dimensions (see Section 6). The motivation is simple: in many ML applications, we want to fit a conservative field (gradient of a potential) such as score-based diffusion models, but in practice, many people have observed that it is easier to work with an unconstrained vector field. By learning the Hodge decomposition, we can effectively remove the non-conservative part of this vector field.
>
> > There are no quantitative comparisons between the two different parameterizations of divergence-free fields (matrix-free and vector-free). This would strengthen the paper.
>
> We have added a new Figure 2 to compare these two approaches, and clarified their differences at the end of Section 2. To summarize, the matrix field is the more flexible approach while the vector field allows us to add in a non-negative constraint by using subharmonic neural networks.
>
> > I am concerned about the applicability for PDE problems. The paper describes avoiding "expensive numerical simulations" by parameterizing solutions with a neural network as a strength, but in truth PINNs are orders of magnitude slower and less accurate than state of the art simulation methods that parameterize solutions on grids. PINNs have indeed received a fair amount of recent attention from the physics community, but are not yet really a practical method, e.g., as demonstrated by the poor results in this paper for modeling the 2D flow shown in Figure 2.
>
> The reviewer’s concerns about the usefulness of PINN solvers are very justified, and we share them. The purpose of our demonstration, however, was more to show that incorporating these constraints into the model at a structural level provides a benefit over existing PINN approaches.
>
> To this end, we have updated the paper with a direct comparison against a “vanila” PINN and our NCL model applied to the fluid problem on the tori. As shown in Figure 4, our model significantly outperforms the penalty approach, and matches well with the reference FEM simulation while the PINN fails to do so. We have also updated our results (we mainly increased network depth and trained for longer) and can show that NCL exhibits _linear_ convergence towards the solution, which we at least think is quite amazing.
>
> That said, we make no claims that our improvements put PINNs on par with FEM for PDE simulations. There has been a lot of research on PINNs due to their inherent adaptability and mesh-free nature, and we have simply brought them closer to the FEM solutions, increasing their reliability.
>
> > The application to the Euler equation with non-constant density is also rather niche.
>
> We did not aim to improve state-of-the-art solving of these PDEs, but simply used this as a test bed to showcase how penalty methods from PINNs are unreliable whereas NCL’s ability to satisfy PDEs structurally can significantly improve the area of neural network-based approximations.
>
> > Any there any applications you can think of that would need more than 4 dimensions?
>
> Oh absolutely. One is handling high-dimensional PDEs; the NCL framework is a “neural PDE” but instead of relying on simulation, we directly enforce the PDE constraint at the architectural level. It is difficult to perform research on high-dim PDEs as there is no reliable solver; however, this approach allows us to sidestep that requirement. Another is simulation-free generative modeling over probability paths. Both applications; however, require us to scale up the NCL framework, which we are excited to tackle. This first work mainly focuses on the structural constraints and universality.
>
> > Can you add quantitative comparisons between the matrix-field and vector-field formulations?
>
> Responded above.

---

> > ### Comment · Reviewer_Mgw4 · 2022-08-09
> > **Thank you for the clarification**
> >
> > The paper has certainly been (marginally) improved by the revision, and I am accordingly raising my score from 4 to 5.
> >
> > I appreciate the compute time comparison in Figure 2, but I would really rather see a comparison of modeling accuracy or a learning curve. Both approaches seem to have roughly similar computational scaling.
> >
> > The new example of the 25-100 dimension vector field is also appreciated, though I'm struggling to understand exactly what this synthetic example is showing. What is the right baseline here? Also, I would appreicate a citation for the claim "many works have reported that it is easier to work with unconstrained vector field"

---

> > > ### Author Response · Authors · 2022-08-09
> > > **Thank you for the reply**
> > >
> > > > I appreciate the compute time comparison in Figure 2, but I would really rather see a comparison of modeling accuracy or a learning curve. Both approaches seem to have roughly similar computational scaling.
> > >
> > > Just to clarify our perspective: the matrix field construction is faster to compute than the vector field (no need to compute a Jacobian matrix, so one less AD operation; Fig 2), and the vector-field construction is essentially just a special case of the matrix field construction (see Fig1). This is discussed when the two constructions are explained, and also now illustrated in the figures, so we didn't bother showing learning curves for both; the matrix-field will simply converge faster in all cases where it is applicable. However, we show that the vector-field construction is still a useful concept since (i) it is universal for divergence-free fields and (ii) it allows us to enforce subharmonic constraints to enforce non-negativity, a necessary constraint for the dynamical OT problem to ensure no mass leaves the system accidentally between times zero and one.
> > >
> > > > The new example of the 25-100 dimension vector field is also appreciated, though I'm struggling to understand exactly what this synthetic example is showing. What is the right baseline here?
> > >
> > > We envision learning the Hodge decomposition to be useful when it is beneficial to remove any divergence-free parts for downstream tasks (e.g. to ensure accurate MCMC sampling). Also, the objective we propose for training is quite interesting too (and is based off of the differential forms derivation). As an alternative we also tried using L2 loss between vector fields but that did not perform nearly as well and resulted in non-convergent behavior. For now, we simply treat this as a short section to showcase scaling behavior and to propose a straightforward application of our framework for learning the Hodge decomposition.
> > >
> > > > Also, I would appreicate a citation for the claim "many works have reported that it is easier to work with unconstrained vector field"
> > >
> > > Ah, actually, all of the cited works in the previous sentence when discussing relevant applications are examples of works that report this in some way, i.e. making use of an arbitrary vector field when the target is a gradient field. This is far from an exhaustive list as well, but we felt these works are representative (e.g. first to propose using arbitrary vector fields for their application, or just first to propose this application). We'll rearrange these sentences a bit to make these citations clearer.

---

### Official Review · Reviewer_RFJW · 2022-07-19

**Rating:** 7
**Confidence:** 2
**Soundness:** 3 good
**Presentation:** 3 good
**Contribution:** 4 excellent

**Summary:**

The authors propose a neural network architecture that can be used to parametrize divergence-free vector fields. As applications, they show how this method can be used to enforce the continuity equation in PDE simulations of incompressible fluid flow, and also used to enforce both continuity and non-negativity when optimizing time-dependent solutions of the dynamical optimal transport problem.

The authors propose two parametrizations in this work, which realize a tradeoff between model size and computational cost.
1. Matrix-field: the vector field $v$ is computed by taking row wise divergences of $n$ matrix fields, ie., $v$ is parametrized by $n$ matrix fields.
2. Vector-field: each matrix field is written in terms of the pointwise Jacobians of $n$ vector fields, ie., $v$ is parametrized by $n$ vector fields at the additional cost of evaluating these Jacobians.

The authors show that, possibly up to a constant vector field, any divergence-free field on the $n$-dim Torus admits a representation in either of these two forms. Lastly, to apply this representation in experiments, the corresponding matrix/vector fields are approximated by trained neural network functions.

**Questions:**

What is a rough estimate of the wall clock time required to generate Figures 1, 2, and 3? In the case of Figure 3, how does this compare to the time required by the baseline method to compute the OT map?


**Limitations:**

The proposed method relies heavily on expensive differentiation operations, such as evaluating the Jacobian of many NNs and evaluating the divergence at points of corresponding matrix field.

**Strengths And Weaknesses:**

The main strength of this work is that the proposed parametrization allows to exactly satisfy the divergence-free constraint, addressing one of the main drawbacks of penalty-based approaches such as in existing PINN methods.

While this is an interesting idea, there are some significant weaknesses in the experimental evaluation:

1. Insufficient quantitative evaluation: for simulating incompressible fluid flow, there is no quantitative evaluation at all of the numerical accuracy of the proposed method. It's not clear from qualitative plots in Figures 1 & 2 whether this method is method is accurate relative to the performance of competing methods. For optimal transport, the authors show comparable performance with one baseline optimal transport solver on toy models. However, since there is also no discussion of the computational cost of NCL, it's highly unclear if it represents a practical contribution to either of these two problem domains.

2. No discussion of computational cost: related to the last point, there is no discussion of the computational cost of either variant of NCL. From the descriptions of the matrix-field and vector-field algorithms, it seems this could be very costly compared to eg. evaluating a feedforward network. Even if this method can achieve quantitative benefits over alternatives, it's not clear whether it can do so in a computationally feasible manner.


_Edit_: I have changed my evaluation of this work based on feedback from the authors and I have raised my score to 7.

---

> ### Author Response · Authors · 2022-08-02
> **Response to Reviewer RFJW**
>
> We thank the reviewer for their candid feedback and for acknowledging the novelty of our approach. In regards to the concerns on quantitative evaluations and computational costs, we have added multiple plots in the revision to address these. Detailed responses to each concern are below.
>
> > Insufficient quantitative evaluation: for simulating incompressible fluid flow, there is no quantitative evaluation at all of the numerical accuracy of the proposed method. It's not clear from qualitative plots in Figures 1 & 2 whether this method is method is [sic] accurate relative to the performance of competing methods.
>
> We had visualized the error but did not label the magnitudes. We have instead changed this into a plot that compares the error against a gold standard FEM solution. In the revised Figure 4, we show that the approach of penalizing the continuity equation does not reliably train the network to recover the solution, even when the training error is very small. In contrast, we exhibit linear convergence towards the solution.
>
> > For optimal transport, the authors show comparable performance with one baseline optimal transport solver on toy models. However, since there is also no discussion of the computational cost of NCL, it's highly unclear if it represents a practical contribution to either of these two problem domains.
>
> We have replaced the original table with a convergence plot with wall clock time on the x axis, and compared against a baseline minimax approach (which is the most alternative efficient parametric method we know of). We show that not only is our approach more accurate than the baseline minimax approach, we can also train faster. We still use the solution from a discrete OT solver as a reference, but we note that (i) it does not provide an optimal map and (ii) scales extremely poorly with sample size, whereas parametric approaches learn a coupling between the full continuous distributions. While we only showed optimal transport in this work, NCL has the potential to be a very versatile generative model.
>
> > No discussion of computational cost: related to the last point, there is no discussion of the computational cost of either variant of NCL. From the descriptions of the matrix-field and vector-field algorithms, it seems this could be very costly compared to eg. evaluating a feedforward network.
>
> Yes NCL is quite a bit more expensive than evaluating a feedforward network—imposing additional structure on the model comes at a cost. This is mainly due to the divergence computations, which we had discussed in the conclusions section. In the revision, we have now added a plot (Figure 2) showing wall clock times for evaluating the two constructions, showing that both scale quadratically with dimension. This current work is mainly focused on the design of structural constraints, while in the future we plan on speeding this up with trace estimation methods.
>
> > Even if this method can achieve quantitative benefits over alternatives, it's not clear whether it can do so in a computationally feasible manner.
>
> We note that there isn’t quite a perfect alternative, since to the best of our knowledge, there is (i) no existing divergence-free construction that is also universal and (ii) there has been no way to exactly satisfy the continuity equation outside of numerical simulation with an ODE solver. That said, we agree with the reviewer that we did not make clear the advantages of our approach. We have now added neural network baselines for both the PDE and OT experiments; where we show we have a very sizeable improvement compared to baselines. Specificially, our method of enforcing the continuity equation structurally can significant improve the line of work in using neural networks to approximate PDE solutions when there is no data.
>
> > What is a rough estimate of the wall clock time required to generate Figures 1, 2, and 3? In the case of Figure 3, how does this compare to the time required by the baseline method to compute the OT map?
>
> The PDE experiments take around 8 hours on our GPU (a single GTX Titan), but as be seen from the plot (Figure 4c), it is still converging. The OT experiments take less than an hour. As mentioned above, the discrete OT solver is only used as a reference and we have added a minimax approach as comparison.

---

> > ### Comment · Reviewer_RFJW · 2022-08-03
> > **Added experiments show a clear contribution**
> >
> > I am happy to see the added experiments and changes to the organization of the body. I feel the paper is both more understandable, and that its new quantitative experiments demonstrate a clear and significant advantage over PINNs in multiple relevant problems. While the method is computationally expensive, the authors remark that trace estimation methods (such as those applied in the recent work of Chen, Duvenaud, and others) are a potential avenue to reduce this expense, indicating that reducing the computational cost while preserving superior accuracy of this method is a new direction opened up by this work.
> >
> > Because of its clear contribution and interesting future directions, I raise my score to 7.
> >
> > Note: there is a broken reference on line 579 of the appendix.

---

### Author Response · Authors · 2022-08-02
**Overall response**

We thank all reviewers for their feedback and constructive comments. We have uploaded a revision to address their concerns. Overall, the revisions consist of rearrangements and more experiments, which have not changed the narrative of the paper but we believe have significantly strengthened it.

Specifically, there were concerns regarding clarity. We have
 - Moved the preliminary on differential forms to the Appendix. This allowed us to streamline the presentation, and gave us more space to add in new experiments to address other concerns.
 - Added more explanations for the scaling issues of our approach (Fig 2), as well as added plots that indicate wall clock time (Fig. 5 & 7). The divergence operation is expensive when naively computed, but we had only discussed this in the conclusion. It is now discussed front and center where we propose divergence-free constructions.
 - Added a Figure 1 to succinctly illustrate our construction.

There were concerns regarding comparisons to baseline methods. We have
 - Added a physics-informed neural network (PINN) baseline to the tori example (Fig 4). We also now plot the quantitative errors instead of just visualizing them (Fig 4c). Amazingly, we show linear convergence to the solution when the continuity equation is enforced structurally, whereas PINN’s training loss is not  indicative of convergence at all.
 - Added a minimax approach to the optimal transport example (Fig 7). This is the fastest alternative approach that we know of for learning W2 optimal maps. Our NCL approach has inherent advantages to this minimax approach, and is a bit faster and more accurate in practice.

There were concerns regarding scaling to higher dimensions. We have
 - Included a new example of learning the Hodge decomposition (Sec 6). This is a useful post-hoc tool for ML applications where the model should ideally be either conservative or divergence-free, but this constraint is not imposed on the model itself. We also show applications to moderate dimensions.

---

### Meta-Review · Area_Chair_MhxP · 2022-08-25

**Recommendation:** Accept
**Confidence:** Certain

**Metareview:**

This paper presents two parameterizations for divergence-free vector fields as outputs of neural networks.  These parameterizations allow for modeling of compressible fluids and Maxwell's equations, as they automatically enforce the continuity equations.  In contrast, existing approaches are based on extra penalty terms and or expensive numerical simulation.  While the approach studied does not beat state of the art finite element approaches, it does provide a significant enhancement for physics inspired neural networks, which will provide a platform for further work to continue advancing neural network approaches for physics problems.

**Award:**

No

---

### Decision · Program_Chairs · 2022-09-14

Accept